‘Mystery big cats’ in the Peruvian Amazon: morphometrics solve a cryptozoological mystery

Naish Darren 1 eotyrannus@gmail.com
Sakamoto Manabu 2
Hocking Peter 3
Sanchez Gustavo 4
1 Ocean and Earth Science, National Oceanography Centre, Southampton, University of Southampton , Southampton , UK
2 School of Earth Sciences, University of Bristol , Bristol , UK
3 Natural History Museum of the University of San Marcus , Lima , Peru
4 Fundación Neotrópico , La Laguna, Tenerife, Canary Islands , Spain
Roberts David
Electronic publication date: 2014 Mar 6
Publication date: 2014
Volume: 2
Electronic Location ID: e291
Received 2013 Sep 24; Accepted 2014 Feb 4
Copyright: © 2014 Naish et al.
Copyright year: 2014
Copyright holder: Naish et al.
License: This is an open access article distributed under the terms of the Creative Commons Attribution License, which permits unrestricted use, distribution, and reproduction in any medium, provided the original author and source are credited.
License URL: https://creativecommons.org/licenses/by/3.0/

Keywords: Jaguar, Morphometrics, Panthera, Felidae, Peru, Cats, Mammals, Amazonia, Skulls

Funding: While we all performed this work while based at our respective institutions, this work was wholly self-funded and we have not received any special funding.

==============================
Two big cat skulls procured from hunters of Yanachaga National Park, Peru, were reported as those of cats informally dubbed the ‘striped tiger’ and ‘anomalous jaguar’. Observations suggested that both skulls were distinct from those of jaguars, associated descriptions of integument did not conform to this species, and it has been implied that both represent members of one or two novel species. We sought to resolve the identity of the skulls using morphometrics. DNA could not be retrieved since both had been boiled as part of the defleshing process. We took 36 cranial and 13 mandibular measurements and added them to a database incorporating nearly 300 specimens of over 30 felid species. Linear discriminant analysis resolved both specimens as part of Panthera onca with high probabilities for cranial and mandibular datasets. Furthermore, the specimens exhibit characters typical of jaguars. If the descriptions of their patterning and pigmentation are accurate, we assume that both individuals were aberrant.

Introduction

The continued existence of undiscovered, large (>10 kg), terrestrial mammal species in the extant fauna is not beyond possibility, despite the expectation that all remaining undiscovered mammals are predominantly small. Indeed, while it is popularly supposed that the inventory of large, terrestrial species is mostly complete, several have been named within recent decades, including the Saola Pseudoryx nghetinhensis (Van Dung et al., 1993), Dingiso Dendrolagus mbaiso (Flannery, Boeadi & Szalay, 1995), Giant or Large-antlered muntjac Muntiacus vuquangensis (Do Tuoc et al., 1994), Small red brocket Mazama bororo (Duarte & Jorge, 1996), Giant peccary Pecari maximus (van Roosmalen et al., 2007; Gongora et al., 2007) and Kabomani tapir Tapirus kabomani (Cozzuol et al., 2013). Indeed, approximately 10% of the 5000 extant mammal species have been named since 1993 (Reeder, Helgen & Wilson, 2007; Ceballos & Ehrlich, 2009), though note that approximately 60% are so-called cryptic species.

Of those large-bodied, terrestrial mammal species named recently, all inhabit tropical forests in southeast Asia, South America and New Guinea. Some or most were known to local people prior to scientific discovery. Furthermore, several were initially known from circumstantial data collected by field researchers. Examples include the Kipunji Rungwecebus kipunji, discovered in 2006 following observations of a mystery monkey (Beckman, 2005; Jones et al., 2005), and Burmese snub-nosed monkey Rhinopithecus strykeri, discovered in 2010 following investigation of local reports about a “monkey with an upturned nose” (Geissmann et al., 2010).

In view of the possible existence of previously undocumented species of new, large terrestrial mammal, and of the recognition of such species following investigation of ethnic knowledge, it is worth taking seriously suggestions that new large mammal species might exist in such regions as tropical South America.

Fieldwork in the Peruvian Amazon reveals that local people refer to several mammals that potentially represent undiscovered taxa (Hocking, 1992; Hocking, 1993–1996). Two large cat skulls, identified by local people as belonging to two of the region’s ‘mystery’ cats, were procured during the 1990s (Figs. 1–2). Both had been defleshed and cleaned by boiling prior to being passed to one of us (PH). In view of this we did not pursue the possibility of extracting DNA (though we recognize the possibility that it still may be possible to extract some using specialized techniques).

Figure 1 ‘Peruvian tiger’ skull, replica of original (CF-0023. Original = MHN 8736).

Specimen shown in (A) left lateral view, (B) dorsal view and (C) ventral view.

Figure 2 ‘Anomalous jaguar’ skull, replica of original (CF-0022. Original = MHN 9397).

Specimen shown in (A) left lateral view, (B) dorsal view and (C) ventral view.

One of these skulls – reported to belong to an animal known as the ‘striped tiger’ – was obtained from a Pasco Province hunter who sold its skin to an unknown party (Fig. 1). The ‘striped tiger’ is allegedly a striped, jaguar-sized cat (Hocking, 1992). We use the less paradoxical term ‘Peruvian tiger’ for this alleged animal. According to eyewitnesses, the body of this animal is mostly reddish and patterned with white, unbranched stripes (Hocking, 1992). A second distinctive big cat skull was obtained in 1993 from another hunter: it reportedly came from a cinnamon-brown and white, leopard-like animal heavily marked with solid black spots (Fig. 2). It was referred to in a previous report as an ‘anomalous jaguar’ (Hocking, 1993–1996).

Large Amazonian cat skulls, conforming to Panthera and not to Puma, are assumed to be those of jaguars. However, preliminary observations indicated that both skulls differed from those of jaguars: the ‘Peruvian tiger’ skull appeared larger and differed in having a shorter, deeper face, a more gracile zygomatic arch and, possibly, a shallower lower jaw. It also differed in proportions: an indisputable jaguar skull exhibited a width:height ratio of 1.59 while the same ratio in the ‘Peruvian tiger’ is 1.37 (Bille, 1997). A convex frontal region appears reminiscent of the same feature in tigers (P. tigris). The possibility that either or both Peruvian skullls might represent tigers is worthy of consideration since anecdotal tales report escaped tigers living in the neotropics (Shuker, 1989). However, in addition to an elevated frontal region, P. tigris possesses long nasals that project beyond the anterior extremities of the maxillae: the ‘Peruvian tiger’ lacks these and appears comparatively short-faced.

We were interested in testing the possibility that either skull might represent a potential new species and therefore subjected both to morphometric tests. High quality casts of the original skulls were used for our study (see Supplemental Information 1) (Figs. 1–2).

Materials and Methods

Thirty-six cranial and thirteen mandibular measurements were taken (see Supplemental Information 1). Since the majority of variance in linear morphometrics reflects an individual’s size, cranial and mandibular morphometrics were standardized through division by the geometric means (GM) of each set (Mosimann, 1970). These measurements were compared to a pre-existing data set compiled by one of us (MS) as part of an integrative analysis of cranial and mandibular morphological and functional evolution in Felidae (Sakamoto & Ruta, 2012).

Linear discriminant analysis (LDA) was conducted on 33 felid species covering 290 specimens for cranial data, and 34 species covering 301 specimens for mandibular data using the same 36 cranial and 13 mandibular standardized morphometrics (Supplemental Information 1). Cranial and mandibular datasets were analysed separately since not all specimens overlap in the completeness of cranial and mandibular data (specimens with missing data were excluded from the analyses so those lacking mandibular measurements would be eliminated in a combined dataset but retained in a cranial only dataset, thus preserving sufficient sample size). Each specimen was given prior classification following recently published taxonomy (Werdelin et al., 2010). The resulting discriminant functions from the training sets were used to predict the classifications of the Peruvian specimens. Since the LDA is performed without the inclusion of the Peruvian specimens, the resulting discriminant functions are unbiased by information from the unknown specimens to be tested. LDA and predictions were performed using the MASS library (Venables & Ripley, 2002) in R (R Core Development Team, 2009) and cross-validated through leave-one-out cross validation (LOOCV).

Although there have been recent criticisms of the use of LDA in morphometric studies, discussions have mostly centered on ordination, visualization of selection gradients, and problems associated with a high number of variables in geometric morphometrics (Mitteroecker & Bookstein, 2011). Since we use LDA in our study for supervised classification, and since we did not use geometric morphometrics but rather traditional measurements, these issues are not pertinent.

Results

LOOCV of LDA on the cranial dataset shows a high percentage of class prediction (94.1%), while that for the mandibular dataset is lower (63.5%). The first three linear discriminants (LD) from the cranial dataset account for 72.3% of the discrimination (LD1, 50.3%; LD2, 13.3%; LD3, 7.66%) and the first eight LDs are required to explain 90% of the between-group variance. For the mandibular dataset the first three LDs only account for 58.7% of the between-group variance (LD1, 31.0%; LD2, 17.1%; LD3, 10.7%) and eight LDs are required to capture 90% of the between-group variance.

Using discriminant functions from the cranial dataset, both Peruvian cats were classified as Panthera onca with posterior probabilities of ∼1. Prediction accuracy is lower in the mandibular dataset, though discriminant functions also classified both specimens as P. onca with posterior probabilities of 0.998 for the ‘anomalous Jaguar’ and 0.961 for the ‘Peruvian tiger’. The predicted positions of the specimens within the LDA-ordinated space for cranial morphometrics are closest to P. onca and P. pardus (Figs. 3–5).

A scatterplot of LD1 and LD2 for large bodied cats (Neofelis, Panthera and Puma) shows the two specimens overlapping with P. pardus (Fig. 3), but a scatterplot of LD1 and LD3 (Fig. 4) shows them plotting closer to P. onca. On the other hand, a scatterplot of LD2 and LD3 (Fig. 5) shows them plotting separately from P. onca and P. pardus.

Figure 3 Scatterplot showing positions of Peruvian specimens relative to other large felids.

This scatterplot depicts the first two linear discriminant axes. These results show both Peruvian specimens to be close to P. onca, but overlapping predominantly with P. pardus.

Figure 4 Scatterplot showing positions of Peruvian specimens relative to other large felids.

This scatterplot depicts the first and third linear discriminant axes. Here, the two Peruvian specimens are especially close to P. onca.

Figure 5 Scatterplot showing positions of Peruvian specimens relative to other large felids.

This scatterplot depicts the second and third linear discriminant axes. In contrast to Figs. 3 and 4, these results show the two Peruvian specimens plotting separately from both P. onca and P. pardus. As explained in the text, a close relationship with P. onca specimens, however, is well supported by morphological characters.

Features consistent with identification of both skulls as those of jaguars include the concave dorsal profile of the nasals, strongly concave lateral surface of the dorsal process of the maxilla (Fig. 6), robust dentition, robust rostrum and relatively broad coronoid process. The two Peruvian specimens exhibit a number of features typical of Panthera including pronounced nuchal and sagittal crests, a short postorbital process and a relatively narrow postorbital constriction (Herrington, 1986). They also possess additional features typical of all Panthera species excepting the snow leopard P. uncia such as a relatively elongate skull, relatively prominent ridges/grooves on the palatal surface posterior to the incisive foramina, and a long palate.

Figure 6 Dorsolateral views of the two Peruvian cat skulls to show jaguar-like characters present in the snout region.

In both specimens, the rostrum and dentition is proportionally robust, the nasals are dorsally convex, and the dorsolateral region of the maxilla is strongly concave. (A) ‘Anomalous jaguar’ skull, replica of original (CF-0022. Original = MHN 9397). (B) ‘Peruvian tiger’ skull, replica of original (CF-0023. Original = MHN 8736).

In view of the superficial similarity with P. tigris, note that the ‘Peruvian tiger’ skull can be distinguished from P. tigris based on smaller adult size (the sutures are still visible but almost fused), a less arched skull profile (the face of the tiger is more downturned), a relatively shorter facial length, and relatively shorter nasals.

Discussion

Our morphometric analysis confirms that both skulls can be identified as those of Jaguar P. onca. Regarding the robustness of our results, lower accuracy of classification of the discriminant functions in the mandibular dataset could be the result of one of two possibilities. Firstly, within-group variance may be high in the mandibular dataset since mandibular morphology may be variable within species. Secondly, the small number (i.e., 13) and/or choice of mandibular variables may not capture species-specific morphology as well as the cranial dataset. Regardless, the fact that the discriminant functions from both datasets consistently assign the Peruvian specimens to Panthera onca with high posterior probabilities places confidence in this assignment.

Despite confident assignment to P. onca, the Peruvian specimens are somewhat morphometrically anomalous, plotting at the periphery of jaguar morphospace in the first three dimensions of LD. However, since at least eight LD axes are necessary to account for 90% of the between-group discrimination, the Peruvian specimens probably only ally with P. onca when a high number of, or all, discriminant functions are taken together, not just the first three.

These results are consistent with the presence of jaguar characters in both specimens, including a concave dorsolateral region on the maxilla and robust rostrum. Ideally, we would corroborate our results with information from postcranial elements and DNA. We urge other researchers collecting ‘mystery’ mammal remains to encourage sources to retain and donate soft tissue samples. If the aberrant coat colours and patterns reported for both cats are accurate (Hocking, 1992; Hocking, 1993–1996), we assume that both individuals were anomalous. Tiger-like stripes and leopard-like solid spots are not typical for P. onca, but similar mutations have been reported anecdotally in jaguar populations elsewhere.

In conclusion, our morphometric analyses indicate that the Peruvian skulls do not represent potential new species. In the case of the ‘Peruvian tiger’, the possibility remains that a jaguar skull was provided in place of the original, however. Should additional cranial material purportedly belonging to mystery big cats be discovered, morphometric techniques such as those employed here should allow their identities to be determined.

Supplemental Information

Supplemental Information 1 Data on whereabouts of specimens and measurements of specimens

Click here for additional data file.

Supplemental Information 2 List of specimens with measurements

List of specimens used in comparisons, with raw cranial measurements (including missing values).

Click here for additional data file.

We thank Ross Barnett, Anthony Giordano and Matt Bille for discussion, Barry Marsh (University of Southampton, UK) for photography, Norman McLeod and an anonymous reviewer for helpful comments and criticisms which improved the manuscript, and the PeerJ team for editorial assistance.

Additional Information and Declarations

Competing Interests

Author Contributions

The authors declare there are no competing interests.

Darren Naish and Manabu Sakamoto conceived and designed the experiments, performed the experiments, analyzed the data, contributed reagents/materials/analysis tools, wrote the paper, prepared figures and/or tables, reviewed drafts of the paper.

Peter Hocking and Gustavo Sanchez analyzed the data, contributed reagents/materials/analysis tools, wrote the paper, reviewed drafts of the paper.

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
