# Peer review of "‘Mystery big cats’ in the Peruvian Amazon: morphometrics solve a cryptozoological mystery"

_PeerJ, doi:10.7717/peerj.291_

## Round 0.1 · original submission · Minor Revisions

An interesting paper. As the reviewers not there is nothing wrong as such with the paper, fig 2 could do with redrafting, but it would be nice, although not essential, if you could address the criticisms of CVA-based approach.

·

Basic reporting

This aspect of the ms is fine. The English style, usage, and grammar adhere to a high standard. The figures need attention, however, the icons, axes, and labels are far too small to be understood easily by readers, either in hard copy or online. These need to be redrafted to make them clearer.

Experimental design

This is somewhat problematic in my view. The question under consideration here is not one of outstanding importance and the findings are largely negative. As written this looks more like a subject that would be suitable for a technical report than a contribution to the scientific research literature. Of course, this does not invalidate the findings which look OK to me. But the experimental design is decidedly 'old fashioned' (we're not talking geometric morphometrics here) and no new ground — either conceptual, mathematical, or biological — is being broken. Perhaps more importantly, the authors have not even gone to the trouble of addressing the criticisms some recent authors have leveled at their CVA-based approach or attempted to confirm the validity of their findings in light of alternative data-analysis procedure suggestions made by these critics (I'm thinking here of the recent 2011 Evolutionary Biology article by Mitteroecker and Bookstein). I believe the authors can make a case for themselves despite these criticisms. But it's clearly not appropriate to ignore them or pretend they don't exist.

Validity of the findings

Despite its limited scope and the negative results this ms would appear to qualify for publication under this heading. I believe the authors could make this into a more impressive and useful contribution to the morphometric literature than they have done date. But according to these criteria there is no reason to reject it outright. But see my comments above.

Additional comments

None that I haven't covered above.

Reviewer 2 ·

Basic reporting

The reference to Table 1 in the text (line 79) is confusing when there is no table associated with the main paper. I have no problem with the measurement system being documented in a supplementary file. However, in the supplementary information file, the caption to the figure showing the measurements states that numbers shown are described in Table 1, which they are not. I would advise adding to Table 1 the numbering system used in the supplementary figure and giving a brief morphological description of each measurement in the table, rather than a long, cumbersome table caption, detailing all of the measurements.

I thought the in-text figures were informative and clear.

Please check the use of commas and semi-colons in lists of in-text citations. Sometimes commas were missing between author names and dates, and sometimes commas separated lists of references rather than semicolons.

Generally the paper was very clearly written. However, in line 93, I found the end of the sentence confusing: perhaps rephrase as “unbiased by information from the unknown specimens to be tested”?

Experimental design

The problem was clearly articulated, the methods employed appropriate, and the findings relevant and of interest. A few minor points require clarification, however. The authors should make clear (lines 83-85) where the comparative dataset came from. Was it collected by them for the purposes of this study, or gained from another source? Also, who took the measurements on the casts? Were there multiple observers and was any test of intra- or inter-observer measurement error undertaken?

Validity of the findings

Results appear statistically sound, clearly described, and the conclusions drawn are robust as a result. In the Discussion, the authors mention intra-specific variability in mandibular morphology as a possible explanation for the lower classification accuracy for the mandibular relative to the cranial data (lines 134-135). It would be nice to see either a reference to a study that has empirically demonstrated this to be the case, or indeed a quick analysis of co-efficient of variation of cranial versus mandibular shape using their comparative dataset would make this point more convincingly.

Additional comments

I found this an elegant, interesting and informative study.

---

## Round 0.2 · accepted · Accept

Looking forward to it being published. Please carefully check the proof.